# Activity Patterns of Captive Red Panda (*Ailurus fulgens*)

**DOI:** 10.3390/ani13050846

**Published:** 2023-02-25

**Authors:** Kathryn A. Bugler, James G. Ross, Adrian M. Paterson

**Affiliations:** Department of Pest Management and Conservation, Lincoln University, Ellesmere Junction Road, P.O. Box 85084, Lincoln 7647, New Zealand

**Keywords:** activity budgets, temperature effects, ex situ management

## Abstract

**Simple Summary:**

We studied how captive red panda (*Ailurus fulgens*) spend their day at three zoos within Australasia, using video cameras, and in-person observations. The red panda in this study were active at dawn and dusk, with another period of activity around midnight. The temperature greatly affected how red panda spent their time, with more focus on resting and sleeping when temperatures were high. This preliminary study suggests environmental changes directly affect red panda behaviour. This knowledge will help inform captive facilities about husbandry and provide insight into how their wild counterparts will cope with global temperatures increasing.

**Abstract:**

We studied the activity budgets of seven *Ailurus fulgens*, at three zoos within Australasia, using video cameras, and in-person observations. Red panda in this study followed a crepuscular activity pattern, with another short peak of activity around midnight. Ambient temperature greatly affected panda activity patterns; red panda allocated more time to resting and sleeping when temperatures increased. This preliminary study suggests how environmental factors affect captive red panda, which will help better inform captive facilities, and how this might impact their wild conspecifics.

## 1. Introduction

Red panda (*Ailurus fulgens*) are currently listed as an endangered species under the IUCN Red List [1] and Appendix I in CITES [2]. Numbers are estimated to be between 10,000 and 15,000, but due to a lack of ongoing monitoring and an elusive nature, these numbers could be as low as a few thousand [1,3]. Red panda are found along the Himalayan Mountain range across Nepal, Bhutan, the northern rim of India, Myanmar, and three Chinese provinces: Sichuan, Yunnan, and Tibet [1,4].

Red panda have proven to be an extremely popular exhibit animal. They are charismatic, easy to work with, and require relatively little floor space, and their primary food source grows almost anywhere [5,6]. In 1977, the international studbook for red panda was created [7]. Individuals that are part of the cooperative worldwide breeding program are accounted for, and their genetics are managed. In 2009, a species survival plan (SSP) was implemented by the AZA (Association of Zoos and Aquariums) small carnivore taxon advisory group (TAG). Captive populations appear strong in number, with ~800 genetically managed individuals worldwide [8], and ~500 unmanaged in China [9]. However, fertility and mortality rates differ significantly across the globe. Mortality rates of cubs in Australasia are around 10%, while in North America they may reach up to 40% [5].

Red panda were long considered to contain two sub-species, *Ailurus fulgens* and *Ailurus styani*/*refulgens*. However, recent genome sequencing by Hu et al. [10] showed that they are actually two species, undergoing a bottleneck after the Penultimate Glaciation (194,000–135,000 years ago). Both species are high-altitude specialists (1400–4800 m) that find temperatures above 24 °C difficult and require protection when mean temperatures exceed 27 °C [1,6,11,12]. Typically, the slightly larger *Ailurus styani* is found at lower altitudes, 1400–3400 m, in the southern Sichuan region of China [4], while the smaller *A. fulgens* is found between 1500 and 4800 m. The majority of *A. fulgens* habitat is found around 3000 m, with bamboo being a key factor for their distribution [13,14].

Wild red panda activity budgets have mainly been studied in Chinese reserves and show a diurnal to crepuscular activity pattern, with less movement around midday [15,16,17]. Yonzon and Hunter [18] studied *A. fulgens* in Nepal and found consistent crepuscular patterns, which may have been affected by human and livestock use of the same areas and resources. Wei and Zhang [19] argue that while it seems that captive red panda follow a more crepuscular lifestyle, this is due to keeper movements and feeding times.

The aim of this study was to assess what factors may affect captive red panda behaviour and activity patterns. As two observational methods were used, we also aimed to see if there are any major differences in behaviours recorded between the two types.

## 2. Materials and Methods

### 2.1. Study Animals

The research was conducted at three zoos in New Zealand and Australia (Table 1). In 2018, 23 zoos within Australasia were part of the Red Panda Species Protection Plan (SPP); 17 of them housed 51 red panda (all captive-born), with the other 5 planning to house red panda in the near future. This study follows 7 out of the 51 panda (13.7% of the population) currently (2018) housed in Australasian zoos.

Two of the panda in this study, Chito and Pasang, were on an anti-inflammatory, once a day for their joints, due to their age.

### 2.2. Observation Methods

During public opening hours, red panda were observed for 1.5 h, twice a day, by the same observer to avoid inter-observer biases. Observation times were varied throughout the day to construct a general picture of daily behaviour. We developed an ethogram (Table 2) and recorded every behavioural occurrence for all panda over 10 min periods. The behaviours were decided upon both before and during observations, which only one person conducted. We modified behaviour descriptions from other red panda studies [20,21]. A definition of scent marking was created following Conover and Gittleman [22]. There was a 5 min break between each 10 min observation period. This was an all occurrences, interval sampling technique, as behaviours could happen at any time during the observations, and the length of behaviours may be cut off by the end of the interval. Also included in the observations was the time of day that the behaviour occurred, current weather conditions, keeper presence, which panda performed the behaviour, and the total length of the behaviour in seconds (to the nearest 10 s).

Video cameras were also used to supplement observations of red panda behaviour. A Blackeye camera (model: BE2-W) (Kinopta, Porirua, New Zealand) was set up outside each enclosure to record behaviour for the entire trial. At Auckland Zoo, this camera was on the enclosure’s back wall, only accessible with a keeper escort, and showed the main latrine site, feeding area, and walkways. Because of the size of the enclosure at Hamilton Zoo, much of it was out of view of the camera, which necessitated a focal area to be chosen. This area encompassed a large tower, many aerial walkways, and a latrine site. The camera was set up on a “staff only” path. The camera could be checked every morning for battery life, and batteries swapped as needed. CWS (Currumbin Wildlife Sanctuary) was also placed on an easy-to-reach perimeter fence so that it could be checked daily. This camera focused on the front platform where the panda slept and where most of the aerial walkways were. Apart from CWS camera footage, which was able to view the entire enclosure and was always recording, all other observations are a proportion of the time spent viewing animals. If animals were off-screen or outside viewing times, these data points were excluded.

This manuscript comes from a larger dataset [23], in which author K. Bugler carried out her master’s thesis. After a period of observations, to gather baseline behaviours, camera traps were put inside enclosures. This is the only point of difference in this study. As camera traps may also affect behaviour, this manuscript only considered the data from before camera traps were placed. Therefore, our observations at CWS were condensed into three days and a week each at Auckland Zoo and Hamilton Zoo. Video cameras were set up to capture as much as possible. Areas included were often sleeping, resting, and main walking areas,

### 2.3. Data Analysis

Time-series graphs (in Microsoft Excel, 2016) were created to show when active behaviours occurred throughout a typical captive red panda day. Chi-square tests for this dataset were carried out in R studio (version 1.3.959) [24].

Models were created using R studio’s “glm” function (from the packages MASS, version 7.3-51.6, and lme4, version 1.1-23). Length of time was the predictor variable, with behaviour as a factor (fixed effects). This formed a basic model, which was then fitted to three alternative generalised linear models to test the best fit: a normal distribution glm, a Poisson glm, and a negative binomial glm. Model fit was assessed using the AIC (Akaike information criterion) function to compare models, and the negative binomial distribution had the best fit. Using the drop1 function on the chosen model, we tested which variables were significant. This test is useful when multiple variables are included in the initial model. Variables are removed one by one from the model and then compared in the output, showing which variable had the least impact on the overall result. Other factors were temperature, date, time, weather, individual panda, and position in the enclosure. These factors were tested, by individual zoo, with the drop1 feature until the model had only significant terms. We then examined how behaviour and other factors affected the response (length of time) on means plots, using the “emmeans” function from the package emmeans (version 1.4.7) [25] ctb (Producer, city, state abbr. if Canada or USA, country), as well as pairwise comparison tests to test for differences between levels of categorical factors.

## 3. Results

### 3.1. Active Behaviours

Behaviours from all panda at all zoos were combined (Figure 1A) and showed a slightly crepuscular pattern of activity, mostly following keeper activity, and another shorter peak around midnight.

Auckland Zoo (Figure 1B) direct observations showed a rise in activity in the afternoon until observations ceased. Compared to the camera footage, red panda were active all through the afternoon and into the early hours of the evening, significantly reducing activity around midnight.

At Hamilton Zoo (Figure 1C), camera footage and observations showed two peaks in activity around 0900–1100 and 1600–1800. Observational data points finished around 1600–1700 due to the zoo opening hours. At CWS (Figure 1D), there was an increase in activity between 0800 and 0900, for both monitoring methods, coinciding with when keepers first arrived with food and clean water. However, the camera footage showed that the morning peak started much earlier, around 0500. There was another spike in activity in the afternoon, when keepers fed the panda again, and activity reduced further from 1830 onwards.

### 3.2. All Behaviours

Sleeping was the most common behaviour at ~68% (observations) and ~44% (camera footage) of the time (Table 3). The next most common behaviours were locomotion (~13%/~22%), resting (~7%/~15%), eating (~6%/~12%), and grooming (~6%/~3%).

There was a significant interaction between the observation method and behaviour type (ꭓ^2^ = 923.56, DF = 17, *p* < 0.001). All three zoos, when analysed separately, showed a positive significant interaction between method of observation and behaviour (Auckland Zoo (ꭓ^2^ = 110.41, DF = 7, *p* < 0.001), Hamilton Zoo (ꭓ^2^ = 232.14, DF = 7, *p* < 0.001), Currumbin Wildlife Sanctuary (ꭓ^2^ = 36.80, DF = 7, *p* < 0.001)). For sleeping, the most common behaviour, there was a significant difference between observational methods (T = 4.541, DF = 190.56, *p* = 9.912 × 10^−6^).

There were overlaps in the duration of a behaviour between zoos (Figure 2), but no overlaps occurred when observational methods were compared (Figure 2). Pairwise comparisons showed differences in mean length (seconds) of behaviours between all zoos (Auckland/Currumbin: *p* < 0.0001) (Auckland/Hamilton: *p* < 0.0001) (Currumbin/Hamilton: *p* < 0.0001). The observational method showed differences in the mean length of behaviours at the different zoos.

### 3.3. Temperature Effects

Daytime temperatures during observations (Table 4) ranged from 11 °C at Hamilton Zoo to 32 °C at CWS. When drop1 chi-square tests were carried out, the duration of behaviours was significantly affected by temperature (Auckland Zoo (ꭓ^2^ = 977.66, DF = 1, *p* < 0.001), Hamilton Zoo (ꭓ^2^ = 914.94, DF = 1, *p* < 0.001), and CWS (Temperature: ꭓ^2^ = 483.11, DF = 1, *p* < 0.01)). Short-duration behaviours, such as scent marking, locomotion, and eating, occurred less often in hotter conditions (Figure 3A–C). Scent marking increased at Auckland Zoo (Figure 3A) between 21 °C and 24 °C, but was not observed at all over 29 °C at CWS (Figure 3C). Locomotion length and frequencies changed with increasing temperatures (Figure 4). More time was given to resting and sleeping (Figure 5) as the temperature increased. All behaviours, and durations of behaviour, were performed at all temperatures at Auckland Zoo and Hamilton Zoo (Figure 3B). The most significant change in behaviours performed was seen at CWS, when temperatures increased above 29 °C.

## 4. Discussion

We estimated captive red pa nda (*A. fulgens*) daily activity patterns and compared them between zoos. This was achieved by monitoring red panda behaviour through in-person observations and a continuously recording camera. When all observations were combined (Figure 1A), there were three peaks of activity (camera footage). The first occurred around 0600, before the zoos opened to the public, and the second occurred at 1700–1800, when zoos closed. The third and shortest peak of activity occurred around midnight. When observational data were considered, we saw two small peaks at 0800 and 1600. A comparison of active behaviours between both monitoring methods revealed that observations alone clearly miss active behaviours, which was consistently the case when zoos were considered independently.

Food presentation across zoos varied and likely affected when animals were active. At Auckland Zoo (Figure 1B), no fruit was left out for panda; instead, keepers would enter the enclosure and wait in the feeding area, offering them “panda cake” (a mixture of specialty biscuits, soaked overnight, with baby food and pear juice). Entering the enclosure to offer grapes was also performed during public-focused encounters. In the afternoon, cut pears were put onto a hanging log with nails so that the panda had to manipulate them off the device to eat. This was the beginning of their peak activity period. Fresh-cut bamboo was always on offer.

Hamilton Zoo (Figure 1C) had less prominent peaks in activity. There was an increase in activity around 0900 when keepers arrived with food and medication for Chito. Cleaning usually took place simultaneously. The fruit was placed in feeding stations that a panda would stand on top of a platform to open. As with all zoos, bamboo was always on offer. The activity was highest between 1500 and 1700, with another short peak between 2200 and 0000.

The most obvious example of panda activity co-occurring with keeper activity was at CWS (Figure 1D). Keepers would first enter the enclosure around 0800 with food and would wait until Pasang approached for his medicine. Cleaning would occur shortly afterwards. Keepers then arrived again around 1400 with another bowl of food that was removed around 1600. Cut bamboo was offered, but the observer never witnessed it being eaten. Pasang was not observed performing any active behaviours between 1130 and 1300, likely due to midday heat.

Previous studies on wild *A. fulgens* show bimodal/crepuscular activity patterns [15,26]. However, more recent studies [27,28] show that a wild red panda’s most active hours are around dawn and midday. Trail camera studies in the wild showed the highest peak of activity between 0400 and 0600, and another between 1000 and 1400 [26]. The previous findings of a crepuscular pattern are consistent with camera footage at CWS. Pasang would sleep from 1900 to the early morning hours when he would scent mark before keepers arrived. Scent marking and other exploratory/territorial behaviours are considered positive indicators of red panda health in captivity [29]. This bimodal pattern of activity during the day clearly followed keeper timings. The activity of captive red panda was influenced by keeper presence and management, only slightly deviating from wild patterns. Care should be taken around feeding times and other keeper activities. Our recommendations are to feed panda early in the day to stimulate natural foraging hours, but to also leave some food overnight in secure devices (so that rodents and birds cannot eat it all). Fresh bamboo should be on offer at all times. If it is drying out, it should be replaced more often and placed in water or cooler areas, as red panda ignore dried-out leaves.

Recent wild studies of *A. fulgens* saw individuals allocating their time to avoid human/animal disturbances and areas with high disturbance [28]. Krebs et al. found that red panda showed anticipatory behaviours before keepers arrived and increased movement once they left [21]. Reid found that Chinese red panda were more active in daylight hours, with decreased movement around midday [16]. Zhang et al. describe two activity peaks at 0700–1000 and 1700–1800 [15]. Red panda were more active during daylight hours, with intermittent crepuscular movements. Johnson et al. found *A. styani* to be more crepuscular and nocturnal than other studies, although the panda studied may not have established a home range [17]. If a majority of red panda time budget studies in captivity are from *A. fulgens*, this may account for the differences in wild studies with *A. styani*.

When comparing the two observational methods, differences in the lengths of behaviours recorded occurred between the types rather than between zoos. Observations recorded longer behaviours (Figure 2). This is likely due to observations being carried out during the day, when red panda spent more time resting and sleeping, which are uninterrupted behaviours.

Red panda in this study were less active (Table 3) than those in wild red panda studies. This is likely due to the increased foraging needs of wild red panda, but it could also be influenced by 2/7 panda in this study being on medication due to age-related illness. Studies of wild red panda observe active activity 45–60% of the time, dependent on season [16,18], whereas our study shows panda to be less active (19–36%). A more recent study [29] found captive red panda at Rotterdam Zoo to be active 40–53% of the time, although the two pairs in the study were young breeding pairs, and perhaps best compared to the pair at Auckland Zoo (Khela and Ramesh). In captivity, panda are brought a selection of food, such as bamboo, fruits, specialised biscuits, and the occasional day-old chick or mouse. This increase in fruit in the diet may lead to more short rests, as the stomach fills more rapidly [16]. This can be seen in wild red panda, which have a high frequency of short rests in spring when fruit and new, more nutritious bamboo shoots are available [15,27]. Glatston (2011) summarises that because “captive diets do not occupy enough of the red panda’s daily activity [7] panda may compensate for this activity ‘vacuum’ by over-grooming and eating hair”. We found that grooming occurred less than in other studies, and no stereotypy over-grooming was observed.

Average temperatures in Nepali/Indian border red panda habitat do not exceed 14 °C [27]. However, because temperatures were only reliably recorded during the daytime for this study (Table 4), we compare the maximum temperatures seen in the wild to this study. Ranges of temperature in Autumn in situ vary from 1.6 to 27.6 °C [27]. In this study, Auckland Zoo and Hamilton Zoo observations took place in Autumn, and temperatures ranged from 11 to 25 °C during the day, putting them in a similar range to the Nepal/India border red panda habitat. Summer ranges at CWS differed from in situ ranges. In the first week of summer daytime temperatures reached 32 °C, at CWS. While the maximum temperature, in-situ, throughout all summer, was only 28.9 °C [27]. Despite small sample sizes, results show interesting trends in red panda behaviour. Having *n* = 1 for CWS only hints at what might be happening for other red panda in this heat. This study indicates a need for further investigation into the impacts of temperature on red panda behaviour, both in captivity and the wild.

The temperature had a significant effect on panda behaviour. The temperature range in New Zealand is more similar to their natural range than the zoos in Australia and much of the rest of the world. At CWS, there were fewer short-duration behaviours (e.g., locomotion) as temperatures (above 29 °C) increased (Figure 3C). Loeffler states that “protection of red panda from temperatures above 27 °C is critical” [11]. The AZA red panda care manual also states that heat stress in red panda is intensified by high humidity [6]. Air-conditioned indoor areas or nest boxes are recommended in these conditions. Misters, which CWS turned on daily, are noted as being a way to provide cooled areas for red panda [6,12]. Auckland Zoo (Figure 3A) and Hamilton Zoo (Figure 4 also had significant temperature interactions, but these interactions were not as obvious to the observer, as even the highest temperature in the New Zealand zoos was not as high as the lowest temperature at CWS. Heat stress can increase stereotyping [30] and resting behaviours seen in captive red panda. Stereotypies may include pacing, tongue flicking, and position circling [31]. Pasang at CWS was seen tongue flicking in the hottest periods.

Temperatures in the upper range are of more concern with red panda than thermal lows, as their natural range rarely exceeds 20 °C. With dense fur coats, a fur covering over their footpads, and an ability to adjust their metabolism, red panda are very well adapted to life in the cold [5]. If red panda are housed in climates that regularly rise above 23.8 °C, in summer, then nest boxes should be in constant shade through the day. If temperatures rise above 26.6 °C, then air-conditioned dens or nest boxes should be provided, especially for pregnant or nursing dams [6]. Noticeable changes in the survivorship of neonates already occur with mean temperatures above 19 °C [5,32]. In contrast, indoor housing or access to an insulated nest box should be provided where winter temperatures drop below −6.6 °C [6]. In areas of extreme cold, supplemental heat should be provided in indoor housing or nest boxes. Preferably, these areas should be maintained between 1.6 °C and 23.8 °C [6]. A reduction in keeper interactions around the hottest part of the day may need to be taken into account to reduce heat stress on days where temperatures are over 27 °C.

Zidar surveyed 69 zoos housing red panda from around the world and found that, on average, summer temperatures ranged between 12 and 36 °C and averaged 23.6 °C [33]. Of the zoos surveyed, 16 had average summer temperatures over 27 °C, of which five had no cooling system in place. This observation is cause for concern as temperature and humidity are consistently attributed to increases in neonatal mortality [5,11,34]. This higher infant mortality is likely due to the change in dam behaviour, where she spends less time in the nest box and more time outside [5,35]. However, the temperature is not the only important factor for juveniles. When combined with high humidity, cubs can suffer from dermatophytosis, leading to necroses of tails and ears [36]. While their native range has high humidity during the summer months, the high altitude keeps summer temperatures low, and therefore, young are not exposed to both factors [5]. If global temperatures continue to rise, captive red panda populations (without heat protection) may become less stable and genetically non-viable [34].

## 5. Conclusions

Peaks in activity occurred early morning, early evening, and midnight. While activity correlated to keeper activity, it also coincided with zoo opening hours and the coolest part of the day. Red panda were more active outside of opening hours and avoided activity during the warmest parts of the day. Care should be taken around when food is presented and keeper interaction/cleaning is occurring.

Research such as this provides husbandry managers with knowledge of the needs of endangered animals under their care. This study suggests that further investigation into which of these factors is the most important may help to improve captive conditions for red panda.

These captive red panda were much less active than their wild counterparts, likely due to differences in diet. Some captive red panda are known to over-groom to fill this extra time; however, this was not seen in our study.

An external thermometer that records temperatures at regular intervals should be added to complement continuous recording, as the temperature was an important factor for time spent moving. Further study on the impact of temperature on red panda activity may help to increase the knowledge for their husbandry manual [6].

## Figures and Tables

**Figure 1 animals-13-00846-f001:**
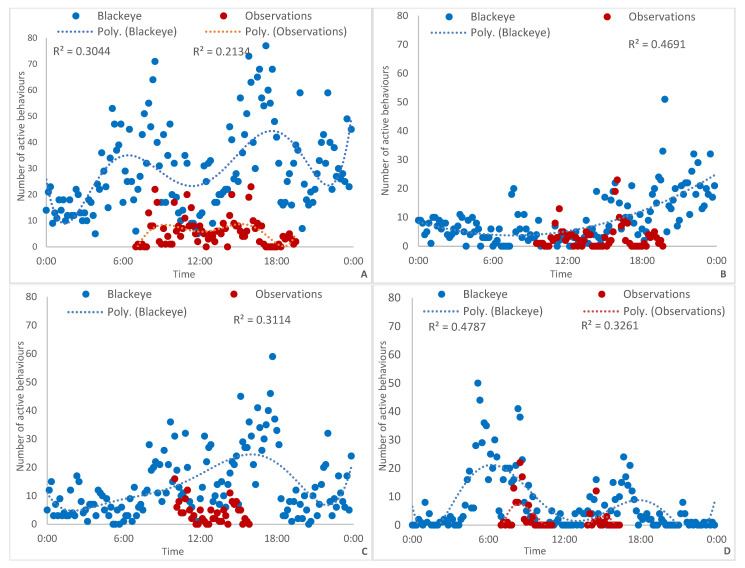
Frequency of active behaviours for red panda at three zoos, in 10 min intervals. The red lines show when direct observations were carried out. R^2^ values are calculated from an nth order polynomial line. (**A**) All zoos combined; (**B**) Auckland Zoo; (**C**) Hamilton Zoo; (**D**) Currumbin Wildlife Sanctuary.

**Figure 2 animals-13-00846-f002:**
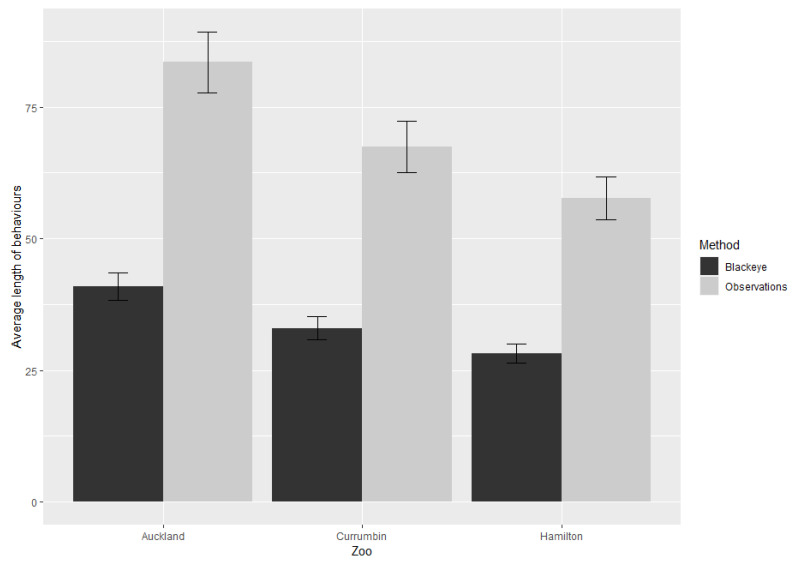
The average duration (response) of behaviours carried out at individual zoos by method type, as determined by Emmeans plot output as a histogram (+95 CI) v 1.4.7.

**Figure 3 animals-13-00846-f003:**
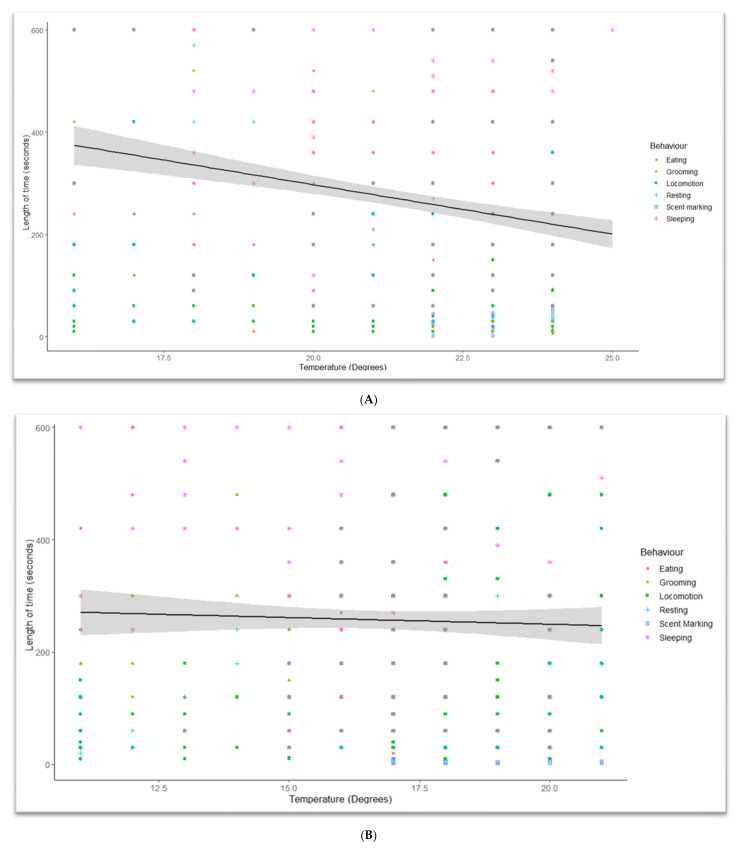
Showing how the duration of behaviours was influenced by temperature at different zoo sites. The five most common behavioural categories were picked for this graph as defecation, interactions, playing, and scent marking occurred infrequently. (**A**) Auckland Zoo, Adj R^2^ = 0.052105, Intercept = 954.88, Slope = −30.654, *p* = 1.7274 × 10^−5^; (**B**) Hamilton Zoo, Adj R^2^ = −0.0012626, Intercept = 264.26, Slope = −0.022011, *p* = 0.99485; (**C**) Currumbin Wildlife Sanctuary, Adj R^2^ = 0.024695, Intercept = −362.83, Slope = 18.718, *p* = 0.0011086. (black line: Estimated means, grey shadow: +95 CI) v 1.4.7.

**Figure 4 animals-13-00846-f004:**
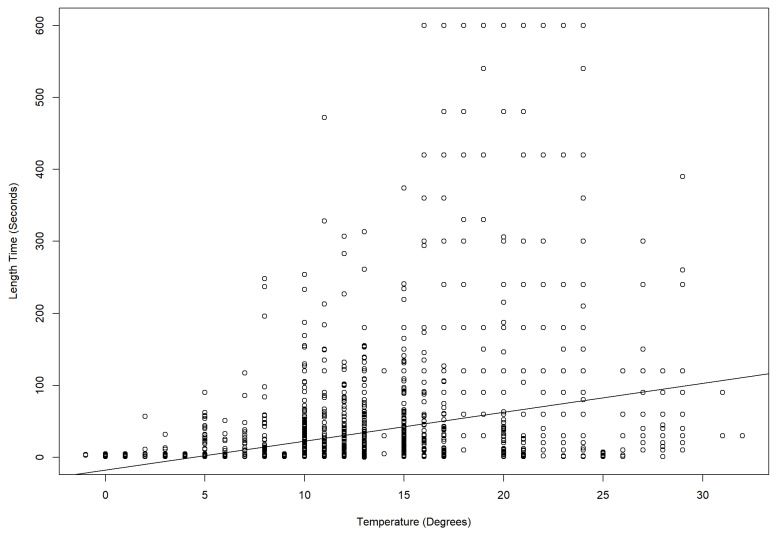
A linear model with a straight line fitted (R studio v 1.4.7) shows that the duration of locomotion events across all zoos was affected by temperature.

**Figure 5 animals-13-00846-f005:**
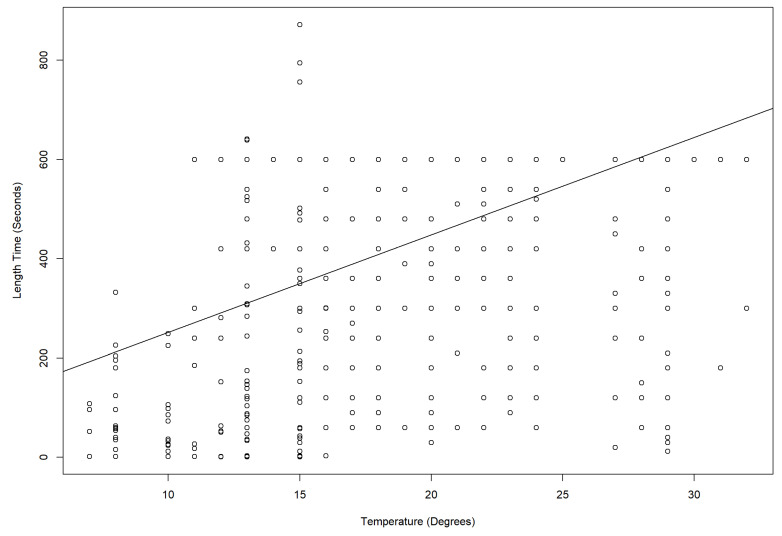
A linear model with a straight line fitted (R studio v 1.4.7) shows that the duration of sleeping events across all zoos was affected by temperature.

**Table 1 animals-13-00846-t001:** Details of red panda studied and the enclosures they lived in.

Zoo	Panda Details	Enclosure Details	Enclosure Size
Auckland Zoo	2.1.0 Ramesh, male, 9 years Khela, female, 6 years Tashi, male, 4 months	Lots of large natural trees, most of which are not native to their natural habitat. Large pond (empty due to juvenile). One main covered platform where feeding occurred. Two nest boxes hidden from public view and an off-display den.	170 m^2^
Hamilton Zoo	1.2.0 Chito, male, 16 years Taylor, female, 10 years Jamuna, female, 4 years	Multiple large exotic tree species, large (empty) pond, a series of off-ground walkways and ramps interconnecting four covered platforms, and a visible large den. One tall man-made tower. A large bamboo stand wrapped around the back half of the perimeter.	529 m^2^
Currumbin Wildlife Sanctuary	1.0.0 Pasang, male, 16 years	Three main platforms at differing heights, interconnected via off-ground walkways. One platform had a roof and the other was fully enclosed, where food and water were given. Natural trees were small. Large off-display den with resting areas and water access. Several water misters.	117 m^2^

**Table 2 animals-13-00846-t002:** Ethogram of red panda behaviours.

Behaviour	Explanation of Behaviour
Locomotion	Movement, on all four paws, that would shift an individual from one area to another, such as by running, walking, or climbing
Resting	Lying, sitting, or standing, eyes open, responsive to surroundings, staying in one area
Sleeping	Lying, curled in a ball or flat, eyes closed, unresponsive to surrounding noise or activity
Eating	Selection and chewing of food brought by keepers, usually fruit, bamboo, or pellets (grass or leaves already in the enclosure may also be consumed)
Drinking	Licking up water in an enclosure, either from an artificial bowl or stream
Grooming/Scratching	Repeated licking or chewing motions of fur or quick paw movements, with claw, across own body
Scent marking	Frequent rubbing of genitals and/or urination on objects around the enclosure
Defecation	Urination, or passing of bowel movement (usually at a latrine site)
Playing	Spontaneous actions that are voluntary and internally motivated. These actions are not associated with the direct need for survival, e.g., eating or predator avoidance. Can be one or more, but is always non-aggressive
Interaction	Between two panda, overall non-aggressive
Keeper interaction	Non-aggressive actions towards a keeper, such as taking food from a keeper or sniffing a keeper’s boots
In den	Out of sight while in the den
Aggressive behaviour	Vocalisation, staring, or aggressive displays towards a conspecific or keeper. Aggressive displays include standing on hind paws with forepaws raised above the head, head bobbing, and slamming forepaws on ground (this behaviour was not seen during the study)

**Table 3 animals-13-00846-t003:** Time spent performing behaviours across zoos as determined by two observation methods. Percentages are calculated from the total time observed/observation type.

Behaviour	Observation	Camera	Zoo
Defecation	0.24%	2.34%	Auckland
	0.14%	5.71%	Hamilton
	0% 0.17%	0% 1.65%	CWS All zoos
Eating	6.84%	28.50%	Auckland
	3.93%	21.66%	Hamilton
	7.86%	0.37%	CWS
	5.89%	12.02%	All zoos
Grooming	4.86%	0.63%	Auckland
	6.9%	1.71%	Hamilton
	4.49%	5.57%	CWS
	5.6%	3.69%	All zoos
Interaction	0%	0.01%	Auckland
	0.33%	0.95%	Hamilton
	0%	0%	CWS
	0.16%	0.16%	All zoos
Locomotion	13.99%	33.04%	Auckland
	11.57%	49.78%	Hamilton
	11.32%	6.63%	CWS
	12.83%	21.66%	All zoos
Playing	0.04%	3.05%	Auckland
	0%	0%	Hamilton
	n/a	n/a	CWS
	0%	0.9%	All zoos
Resting	6.8%	19.66%	Auckland
	2.85%	18.78%	Hamilton
	20.35%	10.75%	CWS
	7.3%	15%	All zoos
Scent marking	0.32%	0.08%	Auckland
	0.21%	0.92%	Hamilton
	0.41%	0.47%	CWS
	0.21%	0.45%	All zoos
Sleeping	66.92%	12.59%	Auckland
	74.27%	0.35%	Hamilton
	41.91%	76.19%	CWS
	67.66%	44.47%	All zoos

**Table 4 animals-13-00846-t004:** Temperatures across facilities during the study period, from observational dataset. Showing only daytime temperatures (0930–1930 at Auckland Zoo, 1000–1600 at Hamilton Zoo, and 0715–1630 at CWS).

Zoo	Average	Minimum	Maximum	Season
Auckland Zoo	22.77 °C	18 °C	25 °C	Early Autumn (March)
Hamilton Zoo	15.5 °C	11 °C	18 °C	Mid-Autumn (April/May)
CWS	27.49 °C	23 °C	32 °C	Early Summer (December)

## Data Availability

Data for this manuscript can be found at: (URL accessed on 21 February 2023) https://docs.google.com/spreadsheets/d/1QFp2r_XDOYeAnLRC5KS0sOADqmM5Y4WA/edit?usp=sharing&ouid=116158200715842018585&rtpof=true&sd=true.

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
