# Peer review of "Activity Patterns of Captive Red Panda (Ailurus fulgens)"

_animals, 2023, doi:10.3390/ani13050846_

Round 1

Reviewer 1 Report

An interesting paper, however some edits are needed to clarify some elements of the study.

Introduction:

The aim at the end of the introduction relates to temperature and how that might affect red panda activity patterns. However, there is little (just lines 44-45) on temperature throughout the introduction. I would like to see some more information on why knowing how temperature affects animal activity patterns is important, and in a captive context, to really support this aim. Otherwise, (and perhaps after reading the rest of the review this may be more suitable) perhaps the aims should be re-structured away from temperature, and more focused on asking what influences captive red panda activity and behaviour, rather than just the effect of temperature.

Materials and methods:

Line 60-62 – these numbers are bit confusing. There’s 18 zoos housing red panda, and this study follows 7 of 18 pandas in 3 zoos? So there’s another 15 zoos housing 11 pandas?

A lot of the panda information in this section could be summarised clearly in a table e.g. “Zoo”, “Panda details” “Enclosure details” “Enclosure size” to make it easy to see and compare without needing to scan through the text.

I would also suggest breaking up this section into smaller headings such as 2.1 Study animals, 2.2 Observation methods, 2.3 Data analysis

Table 1 – Try not to repeat words in the behaviour explanation as in the behaviour category. E.g. eating, does this mean ‘chewing, selection and handling of food, consumption of food, foraging for food’ etc.? See also drinking, grooming, etc.

Line 107 – Do you mean camera traps? If so, what type, settings, etc.? These are mentioned in the abstract but no details are given throughout the manuscript.

Results:

Lines 137-140 – sounds more like methods than results. Suggest moving

Figure 1 – Need to put A, B, C, D on the actual figure so the reader knows if it’s left to right or down for the ordering. Maybe the temperature could also be added so you have the peaks in activity shown with the average temperature change throughout the day? This would help support your arguments later on when talking about temperature and activity patterns.

Figure 2 – I’m not 100% clear on what Figure 2 is trying to show, and if it really is necessary and relates back to your main aim?

Section 3.3 – Could you put in a sentence saying what the temperature range was at each zoo during the study?

I unfortunately struggle with this final (and most important relating to your aim) section of the results. It is stated that behaviours such as eating appeared to occur less often in hotter conditions, but I wonder if this is simply a reflection of their natural behaviour pattern of eating and activity (as mentioned at the start of the results), or is it actually an effect of temperature, or perhaps keeper activity (which didn’t appear to be a variable in the analysis), food availability, or even the number of animals that experienced higher temperatures? It is not clear how temperature was separated as the causal effect here. Peaks in activity around keepers is mentioned in the discussion multiple times, and from what I can infer, it appears the main food is given outside of the peaks of temperature (i.e. early morning and late afternoon), which makes it likely that there would be less eating during hotter temperatures in the middle of the day.

Furthermore, there also needs to be clarification around the statement on line 181-182 and figures 4 and 5. The temperature ranges at the three zoos appear to be different, and so were the number of animals that were exposed to the temperatures accounted for? For example, is there less frequent observations of locomotion (Fig. 4) from 25 degrees above, because there was only n=1 at CWS who experienced these temperatures, compared to n=6 at Auckland and Hamilton who experienced temperatures of <24 degrees? It’s currently not clear if this is taken into account within the analysis and figures.

Discussion:

The discussion needs to relate more about the implications of this specific study, and red pandas in captivity. Should we be careful about what time animals are fed? When keepers are going into enclosures? If there does appear to be increases around keeper activity, should that be monitored to make sure it’s not in the middle of the day to reduce any chance of heat stress, particularly when temperatures get above x degrees. The discussion needs to relate more back to the results of this study and the implications, and not just what other people have reported. The discussion needs to include more information like what is written in the conclusion, with implications of the study. The conclusion shouldn’t be stating new information (this should all be in the discussion), rather summarising the study.  

Author Response

Thank you for your comments! i'll reply to each one as they came up underneath your comment

Introduction:

The aim at the end of the introduction relates to temperature and how that might affect red panda activity patterns. However, there is little (just lines 44-45) on temperature throughout the introduction. I would like to see some more information on why knowing how temperature affects animal activity patterns is important, and in a captive context, to really support this aim. Otherwise, (and perhaps after reading the rest of the review this may be more suitable) perhaps the aims should be re-structured away from temperature, and more focused on asking what influences captive red panda activity and behaviour, rather than just the effect of temperature.

Totally agree and I have re-written the aims

Materials and methods:

Line 60-62 – these numbers are bit confusing. There’s 18 zoos housing red panda, and this study follows 7 of 18 pandas in 3 zoos? So there’s another 15 zoos housing 11 pandas?

Updated with the numbers. Not sure what happened there but I had a second look at the cenus I was sent by the studbook keeper for Australasia before studies were carried out.

A lot of the panda information in this section could be summarised clearly in a table e.g. “Zoo”, “Panda details” “Enclosure details” “Enclosure size” to make it easy to see and compare without needing to scan through the text. I would also suggest breaking up this section into smaller headings such as 2.1 Study animals, 2.2 Observation methods, 2.3 Data analysis

I liked the idea of the table, I have added it in. See if you think the execution was along the lines you were thinking. Also added in smaller headings as well!

Table 1 – Try not to repeat words in the behaviour explanation as in the behaviour category. E.g. eating, does this mean ‘chewing, selection and handling of food, consumption of food, foraging for food’ etc.? See also drinking, grooming, etc.

Few changes on this table as per your recomendations as well

Line 107 – Do you mean camera traps? If so, what type, settings, etc.? These are mentioned in the abstract but no details are given throughout the manuscript.

I've deleted this. Somehow was not picked up by any of the authors, as it was left over from my masters thesis

Results:

Lines 137-140 – sounds more like methods than results. Suggest moving

Moved

Figure 1 – Need to put A, B, C, D on the actual figure so the reader knows if it’s left to right or down for the ordering. Maybe the temperature could also be added so you have the peaks in activity shown with the average temperature change throughout the day? This would help support your arguments later on when talking about temperature and activity patterns.

While I do think adding temperature would help with later arguments, I think the figures would have far too much going on in them if there was a secondary axis for temperatures. I did a draft and there was a lot of points, it was just a bit messy and hard to read. If its deemed a must I can get someone else to take a look, they'll possibly know a better way to do it

Added ABCD to figures

Figure 2 – I’m not 100% clear on what Figure 2 is trying to show, and if it really is necessary and relates back to your main aim?

This was included into my thesis as I wanted to also look at whether or not the observational methods played a part into what types of behaviours are shown. Ive since added extra into the aims and discussion so this relates more. However, it could all be removed if it just adds more questions, rather than answering them!

Section 3.3 – Could you put in a sentence saying what the temperature range was at each zoo during the study?

Added a table with ranges and averages to the results as well as extra written in the discussion 

I unfortunately struggle with this final (and most important relating to your aim) section of the results. It is stated that behaviours such as eating appeared to occur less often in hotter conditions, but I wonder if this is simply a reflection of their natural behaviour pattern of eating and activity (as mentioned at the start of the results), or is it actually an effect of temperature, or perhaps keeper activity (which didn’t appear to be a variable in the analysis), food availability, or even the number of animals that experienced higher temperatures? It is not clear how temperature was separated as the causal effect here. Peaks in activity around keepers is mentioned in the discussion multiple times, and from what I can infer, it appears the main food is given outside of the peaks of temperature (i.e. early morning and late afternoon), which makes it likely that there would be less eating during hotter temperatures in the middle of the day. Furthermore, there also needs to be clarification around the statement on line 181-182 and figures 4 and 5. The temperature ranges at the three zoos appear to be different, and so were the number of animals that were exposed to the temperatures accounted for? For example, is there less frequent observations of locomotion (Fig. 4) from 25 degrees above, because there was only n=1 at CWS who experienced these temperatures, compared to n=6 at Auckland and Hamilton who experienced temperatures of <24 degrees? It’s currently not clear if this is taken into account within the analysis and figures.

Tests used to determine this were outlined in 2.1 data analysis but have added an extra few words here to reiterate the type of test. Drop1 tests showed temperature was significant. This is how it was chosen.

Food is initially offered in the mornings but is available all day to consume.

Discussion:

The discussion needs to relate more about the implications of this specific study, and red pandas in captivity. Should we be careful about what time animals are fed? When keepers are going into enclosures? If there does appear to be increases around keeper activity, should that be monitored to make sure it’s not in the middle of the day to reduce any chance of heat stress, particularly when temperatures get above x degrees. The discussion needs to relate more back to the results of this study and the implications, and not just what other people have reported. The discussion needs to include more information like what is written in the conclusion, with implications of the study. The conclusion shouldn’t be stating new information (this should all be in the discussion), rather summarising the study.

Added in recommendations for reducing keeper interactions in hottest part of days and around feeding times.

Reviewer 2 Report

Comments on Manuscript "Activity patterns of captive red panda (Ailurus fulgens) " Submitted to the Animals

General Comments

I appreciate the opportunity to review this interesting manuscript.

The manuscript deals with an investigation into the activity of captive red pandas in zoos in Australia and New Zealand. Red pandas are an endangered species. The authors used 'in-person' observations and using video cameras. Based on the average duration of behaviors for all individuals, the authors observed that red pandas have 3 activity peaks. The activity pattern of captive red pandas is slightly different from their wild counterparts. The authors explain this dissimilarity due to feeding behavior, which in zoos is predictable and synchronized with the routine of zookeepers.

The manuscript is original and deals with an interesting topic for the management and conservation of red pandas in captivity. The study seems to be well structured, but there are some mistakes and some doubts that need to be considered by the authors.

Simple Summary and Abstract

     The Simple Summary and Abstract must not be identical. The styles of these sections of the manuscript must be different because they serve different purposes. Furthermore, the conclusion of the abstract is not in line with the conclusion of the study. Based on the results observed in this study, it cannot be concluded that behavior in captivity under high temperatures will serve as a model for wild animals. As is well known, captivity has many limitations for an animal to express behaviors and physiological adaptations to climate change compared to the natural environment.

 Keywords

I have doubts about whether the terms “wildlife management” and “anthropogenic” refer to the scope of the manuscript. It is more collateral than central terms.

Introduction

The introduction is well-written.

Material and methods

Time and captivity can change the behavior of animals. It is important to report how long red pandas were in captivity.

Table 1: In the ethogram the description of “playing” is teleological. The description presupposes an individual's internal motivational state, which is not possible. There is also a compliment that it would be a behavior that is not associated with survival, which is an assumption without an empirical basis. The description should be of the individual's movements and the context of the movements.

The authors investigated temperature during direct observations and filming. It is stated between lines 242 and 251 and 254 that wild red pandas modify their activity according to the time of year (seasonality). Therefore, it is important to inform in which season (summer, winter, etc.) the observations took place. The authors cannot neglect that many animal species also synchronize their cycles to the photoperiod.

The methods fail to explain the condition and health of the individuals. Apparently, Chito (line 220) and Pasang (line 227) were being treated for some health disorder. Diseases can affect animal behavior, including activity. Two individuals represent 28.57% of a sample of seven individuals, which is not negligible. The authors must explain how the health of the animals was related to the results.

Results

Line 131: the authors state that all behaviors were combined. However, in a pairwise analysis (line 170), the authors found significantly different mean durations of behavior for red pandas from the 3 zoos. You cannot combine the behaviors of individuals who are different.

Let me give you an example. One researcher supposedly observed the feeding behavior of a lion and an African buffalo. The researcher recorded how much time each of these two species spent eating. While the lion takes 2 hours every other day to feed on a killed prey, the buffalo spends 12 hours eating grass in a day. On average, if the researcher considered that both species are not different, the lion and the buffalo would spend 13 hours [(2 + (12x2)]/2 per day to feed, which is not true. Therefore, the analysis of the combined behaviors does not make clear the activity of the red pandas. As the individuals appear to have some synchrony to the activity in the zoo, it is reasonable to understand that different work regimes in the 3 zoos, can influence the behavior of the animals. I suggest analyzing and discussing the results without the combination of red panda behaviors from the 3 zoos.

In Table 2, it is not clear how the behavior percentages of all zoos were calculated.

Figure 3 has some problems. Figure 3 is composed of three subfigures. It is not possible to know which of the subfigures represents the data of each zoo. The subfigures are not identified (a, b, ...) which zoos the data represent. The scale of temperatures on the X-axis of the three subfigures is different, which makes it difficult to understand the results. The data presented are from 6 behavioral categories, but 9 behavioral categories were observed. Why are the three other behaviors not represented? For a better understanding of the results, I suggest redoing figure 3.

The legend of figures 4 and 5 is missing the statistical values and the type of test.

The paragraph between lines 267 and 274 outlines arguments about other species. The paragraph does not contribute to the full understanding of the text. I suggest deleting this paragraph and its references (27-31).

On line 299 the authors wrote: “Ideally, these areas should be maintained between 1.6 °C and 23.8 °C”. How was it possible to reach this conclusion about the ideal temperature?

Author Response

First of all thank you for your comments. I'll just go through one by one to address them. 

 Keywords

I have doubts about whether the terms “wildlife management” and “anthropogenic” refer to the scope of the manuscript. It is more collateral than central terms.

I have removed these terms

Introduction

The introduction is well-written.

Thank you!

Material and methods

Time and captivity can change the behavior of animals. It is important to report how long red pandas were in captivity.

All red pandas in Australasia are captive born, but have added this in to the mannuscript to avoid confusions

Table 1: In the ethogram the description of “playing” is teleological. The description presupposes an individual's internal motivational state, which is not possible. There is also a compliment that it would be a behavior that is not associated with survival, which is an assumption without an empirical basis. The description should be of the individual's movements and the context of the movements.

I wrote the definition for play behaviour from multiple sources as well as my own observations. So I can always go back and cite them... We can argue that play is or is not directly beneficial for survival but in this context I meant it was not the actions of direct survival. eg eating, predator avoidance etc. I added this into the table to make it more clear what I meant. If I can change the wording then this may help. In the wider context I don’t argue that play is important for survival, espc when juveniles are learning new behaviours!

The authors investigated temperature during direct observations and filming. It is stated between lines 242 and 251 and 254 that wild red pandas modify their activity according to the time of year (seasonality). Therefore, it is important to inform in which season (summer, winter, etc.) the observations took place. The authors cannot neglect that many animal species also synchronize their cycles to the photoperiod.

Added average temperatures for wild and average temps from each zoo

The methods fail to explain the condition and health of the individuals. Apparently, Chito (line 220) and Pasang (line 227) were being treated for some health disorder. Diseases can affect animal behavior, including activity. Two individuals represent 28.57% of a sample of seven individuals, which is not negligible. The authors must explain how the health of the animals was related to the results.

Added into methods that these two panadas were receiving anti-inflammatoies for general aging concerns of joint health. This is also why hamilton had a lot of extra ramps and walkways. And I would say that accenadotaly, these two pandas spent the more time than the others scent marking and walking around. This is probably more sexual diffrences than age related.

Results

Line 131: the authors state that all behaviors were combined. However, in a pairwise analysis (line 170), the authors found significantly different mean durations of behavior for red pandas from the 3 zoos. You cannot combine the behaviors of individuals who are different.

Its to show how all 7 pandas spent their time. When they were more active etc. the zoos are split up. I can do 8 panels having each panda as a graph and then a combine one to show overall, as I was wanting this to compare to how it differs to their wild counterparts. We don’t see individual activity patterns on in-situ studies

In Table 2, it is not clear how the behavior percentages of all zoos were calculated.

Caption added to

Figure 3 has some problems. Figure 3 is composed of three subfigures. It is not possible to know which of the subfigures represents the data of each zoo. The subfigures are not identified (a, b, ...) which zoos the data represent. The scale of temperatures on the X-axis of the three subfigures is different, which makes it difficult to understand the results. The data presented are from 6 behavioral categories, but 9 behavioral categories were observed. Why are the three other behaviors not represented? For a better understanding of the results, I suggest redoing figure 3.

Addressed why only 6/9 behaviours were picked. 3 categories were left out as it muddied the graphs and those behaviours occurred rather infrequently.

Added A, B, C

The legend of figures 4 and 5 is missing the statistical values and the type of test.

Added extra to the figure caption

The paragraph between lines 267 and 274 outlines arguments about other species. The paragraph does not contribute to the full understanding of the text. I suggest deleting this paragraph and its references (27-31).

Was thinking about a broader context, but have deleted.

On line 299 the authors wrote: “Ideally, these areas should be maintained between 1.6 °C and 23.8 °C”. How was it possible to reach this conclusion about the ideal temperature?

Forgotten reference

Reviewer 3 Report

Overall a very interesting paper presenting activity budget information for red panda within a specific zoological region that compared methods of camera trap data to physical in-person observations. The authors could add how the layouts of the various facilities impacted the ability to view active/inactive behaviors since there was an observed difference based on methodology. For in person observations, the authors should address how inter-observer reliability was achieved. 

Author Response

I've added a comment in the methodology saying that there was only one observer. 

I look forward to more comments from this reviewer when they look at the second submission

Round 2

Reviewer 1 Report

Thank you for addressing most of my comments, particularly the addition of tables to make the manuscript clearer. 

A few minor things:

Table 1 and Line 172 - Hamilton spelled wrong

Line 54 - activity spelled wrong

Line 98 and 101 - manuscript spelled wrong

Line 53 - I think this should be 'affect'

Figure captions 4 and 5 - bold half way through the caption

Trail cameras still mentioned in line 112, but previously (line 101) mentioned that data is from before camera traps in place. 

Table 4 - missing degrees C for the Auckland average temperature

Figure 3 - x axis is different between the three panels making comparisons difficult

Other comments

I think there are some factors missing in the model that may (or may not) affect the results. Included in the model was temperature, date, weather, individual, and position. What about time, and keeper presence (or time since keeper presence)? This is a big discussion point, so could it have been included in the model to see if it did affect their behaviours. Were the three zoos analysed separately during the drop1 function? This is not currently clear in the methods whether this was all data combined, or done separately for each zoo. 

I'm still hesitant at the inclusion of the CWS panda, particularly where the data is combined for all three zoos. It is n=1, for less than half the time of the other zoos, in a completely different climate, season, and temperature range, and this could be skewing results. I suggest seriously thinking about the inclusion of this data, and what it means. If it is still included, then the limitations and possibly effects of this need to be included in the discussion.

Panels A and C in Figure 3 show quite contrasting results, yet there is little discussion around this and why this could be/what this means. 

Author Response

Authors' reply: Adjusted the minor things

I think there are some factors missing in the model that may (or may not) affect the results. Included in the model was temperature, date, weather, individual, and position. What about time, and keeper presence (or time since keeper presence)? This is a big discussion point, so could it have been included in the model to see if it did affect their behaviours. Were the three zoos analysed separately during the drop1 function? This is not currently clear in the methods whether this was all data combined, or done separately for each zoo. 

Authors' reply: Added to methods to state that it was done zoo by zoo..

Time of day was actually looked at in very early models but ultimately left out because it was never significant We could have looked at time to keeper presence and whole range of other factors but we are just reporting on the ones we did measure. There is another paper out there that did measure anticipatory behaviours in red panda and gorilla in regards to keeper presence but it was not our aim to replicate that study.

I'm still hesitant at the inclusion of the CWS panda, particularly where the data is combined for all three zoos. It is n=1, for less than half the time of the other zoos, in a completely different climate, season, and temperature range, and this could be skewing results. I suggest seriously thinking about the inclusion of this data, and what it means. If it is still included, then the limitations and possibly effects of this need to be included in the discussion.

Authors' reply: Added to discussion 303-307

Panels A and C in Figure 3 show quite contrasting results, yet there is little discussion around this and why this could be/what this means. 

Authors' reply: Although there is almost no overlap on the x axis so that may explain the response. The axis have been adjusted so that the graphs show the same. This may make it a little more clear. When the temp is in the early 20s then as it gets warmer there is a reasonable diversity of behaviours but they reduce in duration to 200 secs and stay the same till around 29C. After that the pandas are mainly sleeping and resting for long periods.  

Reviewer 2 Report

Dear authors

I appreciate the opportunity to review your manuscript. The authors modified substantial parts of the article, improving the presentation of the text. However, there are still some parts of the text that are not good.

The authors continue to barely define what play is in red pandas. With the description in Table 1, it is very difficult for another researcher to make reliable observations about a play like the authors of this article. You should describe the structure, i.e. the topography of the red panda’s play behavior, and the context of the play behavior. I suggest the author see the definition in Burghardt (2005).

In two red pandas, there are health problems, namely bone joint problems, according to the authors. The animals receive anti-inflammatory drugs. Both pain and anti-inflammatories can alter locomotion in animals, therefore the discussion of the results should include the health condition and treatment of red pandas as factors that influence the results.

I disagree that the author can combine the behaviors of a few animals that are different. The author's argument is not convincing.

“Addressed why only 6/9 behaviours were picked. 3 categories were left out as it muddied the graphs and those behaviours occurred rather infrequently.” This argument is not valid to exclude playing, defecation and interaction. In fact, it is contradictory because the index of Scent marking behavior was low infrequently as well (0.45%).

Reference

Burghardt, G. M. 2005. The Genesis of Animal Play: Testing the Limits. Cambridge,

Massachusetts: MIT Press.

Author Response

The authors continue to barely define what play is in red pandas. With the description in Table 1, it is very difficult for another researcher to make reliable observations about a play like the authors of this article. You should describe the structure, i.e. the topography of the red panda’s play behavior, and the context of the play behavior. I suggest the author see the definition in Burghardt (2005).

Authors' reply:  So I dont delay getting back any longer, I cant address this comment right now. I cant get access to this book at the moment and im just working out access with another

In two red pandas, there are health problems, namely bone joint problems, according to the authors. The animals receive anti-inflammatory drugs. Both pain and anti-inflammatories can alter locomotion in animals, therefore the discussion of the results should include the health condition and treatment of red pandas as factors that influence the results.

Authors' reply: Added line 281. 

Anecdotally, these were probably two of the more active panda. Chito at Hamilton certainly spent more time during observation walking around the enclosure than Taylor. 

“Addressed why only 6/9 behaviours were picked. 3 categories were left out as it muddied the graphs and those behaviours occurred rather infrequently.” This argument is not valid to exclude playing, defecation and interaction. In fact, it is contradictory because the index of Scent marking behavior was low infrequently as well (0.45%).

Authors' reply: Scent marking has been removed from graphs to show the 5 most common behaviours as scent marking, although frequent, only occurs within a 10 second window and therefore length of time doesn’t change because of temperatures, but just the frequency

Updated graphs to have the same axis. It does make it hard to see close up as the ranges were large, have included a combine graph for reference, but wont include that for the feedback given so far.